# Is Dietary (Food) Supplement Intake Reported in European National Nutrition Surveys?

**DOI:** 10.3390/nu15245090

**Published:** 2023-12-13

**Authors:** Eleni Maria Papatesta, Anastasia Kanellou, Eleni Peppa, Antonia Trichopoulou

**Affiliations:** 1Center for Public Health Research and Education, Academy of Athens, Alexandroupoleos 23, 11528 Athens, Greece; m.papatesta@hhf-greece.gr (E.M.P.); akanellou@uniwa.gr (A.K.); e.peppa@hhf-greece.gr (E.P.); 2Department of Food Science and Technology, University of West Attica (UniWA), 28 Agiou Spyridonos Str., 12243 Egaleo-Athens, Greece; 3School of Medicine, National and Kapodistrian University of Athens, 75 Mikras Asias Str., 11527 Athens, Greece

**Keywords:** dietary supplements, food supplements, national nutrition surveys, micronutrients, European countries

## Abstract

Dietary (food) supplements (DSs) have seen a sharp increase in use and popularity in recent years. Information on DS consumption is vital for national nutrition monitoring. The objective of this study was to investigate whether DS intake was reported in the National Nutrition Surveys (NNSs) in all European countries. NNSs reporting DS use were retrieved via literature review (i.e., PubMed, Google Scholar, Scopus), scientific and organizational publications (EFSA), or open-published government and other official reports. Included were the European NNSs referring to adults, published in English, French, or German, post-2000. Out of the 53 European countries, 30 recorded DS intake. Among them, related findings on the percentage of DS intake were published in 21 cases, 5 of them written in the local language. DS use varied by nation, with Finland and Denmark having the greatest (over 50%) and Italy having the lowest percentage (5%). In terms of comprehensive reported data on DS consumption in Europe and the investigation of the contribution of DSs to total nutrient intake, there is a need for improvement. Common DS categories should be defined upon agreement among the involved scientific parties to allow for comparable data and estimations between surveys.

## 1. Introduction

In 1994, the US Food and Drug Administration’s (FDA) definition of a dietary supplement (DS) was included in the Dietary Supplement Health and Education Act (DSHEA) of 1994. According to DSHEA, “The legislation creates a new category of food by specifically defining dietary supplements to include the following “dietary ingredients”: vitamins, minerals, herbs or other botanicals, amino acids or other dietary substance(s) for use by man to supplement the diet” [1]. Similarly, according to the definition by the European Food Safety Authority (EFSA), “Food supplements are concentrated sources of nutrients (i.e., minerals and vitamins) or other substances with a nutritional or physiological effect that are marketed in “dose” form (e.g., pills, tablets, capsules, liquids in measured doses)” [2]. In the European Union, food supplements are regulated as food by the European Commission under various directives but mainly Directive 2002/46/EC of the European Parliament and the Council [3]. Prominent differences are observed between different countries regarding not only the definition of DSs but also legislation and regulation [4]. The same product can be regarded as food in one country and as medicine or DS in another [5]. DSs are not meant to be used for disease prevention or therapy, and they lack the necessary properties to do so. Additionally, they should not be used as a replacement for a balanced diet [2,3].

Since 2018, the DSs market has been expanding, and it is anticipated to do so until 2028, when it is estimated to be worth USD 239.4 billion, compared to about USD 137 billion in 2021 [6]. This augmentation may be attributed to their easy accessibility since they are sold in supermarkets and convenience stores and are over-the-counter products. Furthermore, regulations and laws regarding DSs are rather loose compared to drugs. No pre-market approval is needed, and manufacturers are not obliged to demonstrate the safety and efficacy of the supplement [5,7]. 

Routine supplementation is not advised for the general public, but supplements might be necessary for certain high-risk groups. The recommendations for DSs include folic acid for women of reproductive age before conception or pregnant women in their first trimester to prevent neural tube defects; supplements such as iron and vitamin D for the increased needs of pregnant women; vitamin D for breastfeeding infants; and B12 for the elderly due to lower absorption [8]. A healthy and balanced diet is enough for the acquisition of the necessary micronutrients, but in some situations, DSs might be helpful when it is impossible to meet nutritional needs by diet alone [9].

Official nutrition guidelines are expressed in terms of foods. Consequently, the nutrient-oriented supplement does not warrant the quality of the dietary pattern. Current public health messages for health promotion do not advocate supplements in optimally nourished populations [10]. On the other hand, there are DS supporters, such as “Food Supplements Europe”, who have the opinion to integrate DS use into nutritional policies [11]. Given current concerns about DS consumption [12] in terms of them not yielding the expected health benefits [13] or even harming health [14], not recording DS intake is an important knowledge gap. This implies that nutrition policies without the recording of DS intakes in their National Nutrition Surveys (NNSs) are based on limited data, which is of concern as DS consumption has been exponentially rising [6] in most countries in recent decades. This highlights the need to clarify major barriers and work with countries to establish mechanisms to overcome them and subsequently devise and implement NNSs, including DS records.

Nutrition surveys have an important role in the dietary assessment of the whole population. Separately or together with health surveys, they form the main source of data regarding dietary risk factors worldwide across decades [15]. Findings of a review of available NNS (post-1990) within WHO Europe’s responsibility [16] indicate that only 64% are recording the whole diet data of individuals. The goal was to further explore which NNSs are recording the intake of DSs. This type of survey can identify areas of concern and thus indicate the need to promote best practices across countries [17]. A review of total food intakes, DSs included, is needed to identify where in Europe modification of the diet is necessary. This paper highlights the existing gaps in nutrition surveys regarding DS intake in the context of whole diets at an individual level. To our knowledge, this is the first paper aiming to document and review DS consumption in European countries.

## 2. Materials and Methods

### 2.1. Study Selection

For the current approach, to identify the DS consumption data in European countries, NNSs were retrieved, and a search for the record of DSs in national representative studies was also conducted. Several approaches were used to retrieve and explore the reports or publications. 

A systematic literature review was conducted using Web of Science, Medline/PubMed, Google Scholar, and Scopus to detect academic papers using various search terms (“Dietary supplement” or “Food supplement” or “Nutritional Supplements”, “National health surveys”, “National Nutrition surveys”, “Dietary surveys”). A search was conducted for each of the 53 European countries of the WHO European Region [16].

General web-based searches were also conducted, i.e., Internet searches using the above terms since the survey results may be published as government or other official reports rather than academic papers. Checking the reliability of the sources detected, which was undertaken upon the identification process, a potential bias could be the non-identification of existing related documents uploaded in not easily detectable places at the site of an organization. Any non-open access official reports could apparently not be included in the current study.

We finally searched for studies included in the European Food Safety Authority (EFSA) database [18], The Global Dietary Database (GDD) [19] and the review paper by Rippin et al. regarding National Nutrition Surveys of the WHO European Region [16].

The inclusion criteria comprised European nationally representative surveys that were conducted post-2000, at an individual level, that included male and female adults of all age groups, based on whole diets (not limited to particular food groups), with their results published in academic journals, official reports, and reliable websites. In the case of frequently running NNSs with multiple collection dates, the most recent was selected that had information regarding DS usage. Additionally, any text referring to DS records (or a similar meaning, such as “food supplement”) was identified and retrieved from the survey publication. We included studies that were written in English or provided an English summary or table of the available findings in French or German, while outcomes that were published in another local language were excluded from this comparative approach. The authors were fluent in the languages mentioned above; therefore, related texts could be interpreted and thus included in the current study. It was decided to avoid eventual misinterpretations by using automated translation tools for other EU languages. In case the survey population involved all age groups (inclusion of children and adolescents), the information for this group was not left out, but surveys focusing only on non-adults were excluded. Surveys including all age groups could somehow bias the outcome of the current study when the sample population included children/adolescents or was limited to people of older age. NNSs that included only women and children were also excluded.

### 2.2. Methodologies of Recording Dietary Supplement Use in Different Countries

Several studies have been published regarding DS use. As expected, different DS assessment tools were applied to evaluate DS use, including completing 24 h recalls (one or two), 48 h recalls, Food Frequency Questionnaire (FFQ) from in-person interviews by mail or telephone, and 3- or 4-day food diaries. The time period of consumption differed among the studies. A DS user was someone who had consumed a certain DS in the previous 24 h, the previous days or the previous year. The age groups also varied in different studies.

By reviewing the existing publications, it was observed that the definitions and the categories of DSs varied a lot among surveys. In some studies, DS also included medication with the addition of vitamins. Some studies had information only regarding vitamin and mineral supplements. There is no global consensus on defining the various DSs, natural health products, or complementary medicines in different countries [20,21]. The statistical evaluation of the data differed between studies. Because of the different methodologies used, it is not considered easy to compare the results of the various studies.

DS data that met the inclusion criteria mentioned above were retrieved, and the findings were compared with the limitations of methodological varieties.

## 3. Results

Upon editing, the final material is presented in Figure 1.

From the 53 European countries investigated, information regarding the surveys is listed in Table 1.

### 3.1. Figures on DS Use

The highest DS consumption was found in Finland [22] and Denmark [23], where more than half of the adult population were DS users (DSUs). Similar figures were recorded among Switzerland (47%) [24] and the Netherlands (42%) [25], followed by 38.3% in Belgium [26], 31% in Greece [27], 26.6% in Portugal [28], 25% in Hungary [29] and 24.3% in Germany [30]. France (21.8%) [31] and Sweden (21%) [32] had fewer DSU in the total population. Finally, Spain (13.3%) [33], Poland (10%) [34] and Italy (5%) [35] had the lowest DSU percent. In the UK, 22% of 19–64-years-old and 41% of the over 65-years-old group [36], and in Ireland [37], 28% of the 18–64-years-old and 37% of the over 65 years old group were DSU. The studies that included data on DS consumption are listed in detail in Table 2.

Comparing DS use in the European countries with the rest of the world, indicatively, the USA presents high DS consumption (52%) [38], followed by 45.96% in Korea [39], 45.6% in Canada [40] and 43.2% in Australia [41]. China presents with 0.71%, showing an impressively low DS consumption [42].

### 3.2. DS Preferences by Gender and Age Groups

In all countries, women presented higher DS consumption compared to men. The highest percentage of women DSU was noted in Finland (64%) [22], whereas, for men, it was noted in Denmark (51%) [23]. Table 3 includes percentages of DS users.

DS preferences by age group were also recorded in some studies. Different trends were noted among the different countries. In the UK, Ireland, the Netherlands, Finland, Belgium, Germany and Switzerland, the older population consumed DSs at a higher percentage compared to the younger one [22,24,25,26,30,36,37]. In contrast to these countries in Greece, France, Italy and Poland, generally, there were more younger adult DS consumers than older ones [27,31,34,35]. The percentages between the younger and older populations in Portugal, Spain, and Hungary were alike or had little differences [28,29,33]. 

### 3.3. DSs by Education

The association between education and DS consumption was explored in several studies [22,24,26,27,30,31,34]. The results were similar among these different studies, i.e., DS consumption is positively associated with the level of education, except in Greece, where DS consumption was higher among individuals with an intermediate level of education, in Poland, where subjects with upper secondary education were more often DSUs, and in Denmark, where no differences were documented.

### 3.4. Number of DSs Consumed

In Belgium [26], during the 24 h recall for DSU, it was noted that 70.7% consumed one type of supplement, while 17.8% consumed two, 5.2% consumed three and 6.3% consumed more than three different types of supplements. In France [31], for the previous year, the corresponding numbers among DSU were 51%, 29%, 11% and 10%. In Germany [30], 62% reported using one type of supplement, while 26% reported using two. In Poland [34], 76% reported one type of DS, 18% two, and 1–2% three or more.

### 3.5. Season of DS Consumption

It is noted that in the Netherlands, the number of DSUs increases during the winter months [25]. The same applies to French people, who use DS more in the winter [31]. In Belgium, there are slightly more DSUs during the winter, but this is not statistically significant [26].

### 3.6. BMI–Physical Activity–Health Status

In Greece, overweight men had higher DS consumption, followed by obese and normal-weight men [27]. In Belgium, the results changed according to the type of questionnaire considered. According to the FFQ, DSUs were more common among Belgians of normal weight than among those who were obese. (40.4% vs. 30.9%). On the contrary, taking into account answers from 24 h recall, DSUs were more prevalent among obese (75.2%) compared to normal weight (52.4%) and overweight (40.7%) [26]. Physical activity revealed no link to DS use that was statistically significant in Greece and Denmark [23,27].

In Greece, men who self-reported their health status as “very good” or “good” presented with a high percentage of DS use [27]. In Poland, higher DS usage was noted among participants who stated having “neither good nor bad” or “good” health status [34]. In Germany, poor health status was linked to higher DS consumption [30], while in Denmark, perceived health status was not significantly associated with DS use [23].

### 3.7. Types of DSs Preferred

In Greece, the preferred DS categories were multivitamins with minerals (MVM), multivitamins (MV), calcium, and iron, evenly distributed with around 5% each [27]. In Italy, MVM supplements were also the most frequently consumed supplements, followed by single vitamins or minerals, herbs and botanicals [35]. The most preferred supplements by the Portuguese population were minerals (36.3%) or MV (36.2%) [28]. In Ireland, 30% consumed MVM, 20% fish oils, 12% MV, 11% single vitamins and 8% single minerals [37]. In the UK, responders preferred fish oils and MV with or without minerals [36]. In Belgium, vitamin D was the most preferred supplement (19.2%), followed by MVM and vitamin C [26]. Dutch individuals tended to consume vitamin D-containing supplements more frequently (especially the elder ones), followed by MVM and vitamin C [25], while in Germany, the most frequently used supplements were MVM [30]. The Swedish consumed mostly DSs with omega-3-fish oil (26%) and MVM (24%) [32]. In Finland, the most preferred DSs were vitamins and MVM, but 19% of women and 15% of men also preferred fatty acids [22]. More specifically, for both sexes, the highest percentage concerned vitamin D-containing supplements, followed by vitamin E, magnesium, and vitamin C. Polish people preferred the consumption of vitamin supplements the most, and more specifically, the vitamins most frequently used were B6, Vit C and Vit D, whereas the most common minerals were magnesium, zinc, and iron [34]. In Switzerland, 30% preferred minerals, compared to combined products and vitamins [24].

### 3.8. Type of DS According to Age

In the study from Belgium, younger adults (18–39 years old) frequently consumed omega-3 fatty acids, iron, and MVM supplements, whereas older adults (40–64 years old) consumed vitamin D and calcium supplements [26]. In Germany, vitamin D and folic acid supplements were preferred among older adults. Surprisingly, under 10% of women of reproductive age consume folic acid supplements [30]. In Greece, MVMs were preferred by the younger men in the 18–34-year-old group, while iron was preferred by the over the 65-year-old group. On the contrary, women 18–44 years old were iron supplement users in a greater percentage compared to the older ones, and vitamin D and calcium supplements were not used by women under 44 years old as often as the older groups [27]. The age group behavior in Portugal was mixed, where the younger age group (defined as 18–64 years), compared to the older group (comprising 64–85 years), was more concerned with vitamins and MV consumption, while mineral consumption was much higher in the older group [28]. In Switzerland, a greater percentage of the 65–74 age group consumed mineral supplements by a considerable margin [24], while in the UK, cod liver and other fish oils were most frequently used by the older age group, over 64 years old [36]. Finally, the Dutch population, especially older women over 50 years old consumed a greater percentage of vitamin D and calcium supplements in contrast to iron, which was consumed in a greater percentage in the younger group [25].

### 3.9. Contributions of DSs to the Diet

Few studies have explored the link between diet and DS contribution to vitamin and mineral intake. Indicatively, in Italy, when only supplement users were concerned, supplement supply to overall intake was remarkable for vitamin D (54%), vitamin E (45%), vitamin B12 (42%), and iron (34%). When the study included both DS and non-DS users, the contribution to overall consumption was minimal and, more precisely, less than 9% for all vitamins and minerals [35]. In Ireland, vitamin D supplements contributed to the daily intake of the corresponding vitamin by 9% for the 18–64 age group and 17% for the over-65 age group, while for vitamin C, it was 9% for both age groups, and for calcium, 2% for the 18–64 age group and 7% for the older age group [37]. In the Netherlands, DS containing folic acid contributed 53% to the intake of folic acid (and 11% of folate equivalents), while for Vit D, the corresponding number was 15%. Vitamin C, vitamin B1 and vitamin B2 contributed 10%, while the percentages for other vitamins were less [25]. In Belgium, the biggest contribution concerned vitamin D at about 6%, while vitamin C was only at about 1% [26]. 

In Finland, the amount of vitamins and minerals in the diet of supplement users was at least equal to that of those who were not consuming DSs [22], while in Denmark, the two groups did not present differences regarding intake, except for the age group 18–49, where DS users presented higher intake for most micronutrients. Danish people who had a healthier diet were more likely to be DS users [23]. In Greece, higher consumption of fruits had a positive correlation with DS use [27]. That was also the case in Germany, where DS users consumed more fruits and nonalcoholic drinks and less meat, and it seems that supplement users followed a more balanced meal selection [30]. In the UK, people who took supplements consumed more vitamins and minerals from nutrition than people who did not. The input from supplements had a minimal impact on the percentage of participants who fell below the lowest reference nutrient intake (LRNI) [36].

For most nutrients, the diet reference values (DRVs) were attained or surpassed by DS users. For example, in Germany, with the exception of vitamin D and folic acid, whose intake was much lower, the majority of vitamins were consumed in accordance with the DRVs for nutrient intake [30]. The same applied to minerals with the exception of iron, especially for women of childbearing age; this was also the case with calcium. That led to a great augmentation when the intake of supplements was added, i.e., for vitamins group B, vitamin C, and niacin, intake from the diet and DSs was double the DRVs. In Poland, some study participants consumed far more vitamins than the DRVs, including biotin, vitamin B12, vitamin C, riboflavin, niacin, and vitamin B6. Additionally, the consumption of vitamin B6, Vitamin D, or magnesium by some participants was on par with or greater than the tolerable upper intake level (UL) [34]. In Finland, a portion of the population’s typical nutritional requirements for vitamin A, thiamin, folate, and among men, also for vitamin C and riboflavin, were not met when both diet and DSs were taken into account, while the UL was exceeded concerning the intake of pyridoxine, zinc, and vitamin D [22].

## 4. Discussion

Of the 53 European countries studies, 33 had national nutrition surveys conducted. Although 30 countries describe their methodology record of DS consumption, only 21 of them have published their findings regarding DS consumption, and 5 of them published their results in the local language.

Most countries included in our study originate from Western European countries, whereas Central and East European countries are underrepresented since published data from these countries are limited. In terms of general conclusions regarding DS use in Europe, this should be considered. It is worth mentioning that in 38% of the European countries, information regarding NNSs referring to the whole population could not be found. Since NNSs are necessary for tracking the nutritional status of the population, it is vital for more countries to conduct them. Moreover, in nine countries for which DS consumption data were recorded, the findings were not published; thus, they should be encouraged to publish their results. Furthermore, the five countries publishing NNS data in their local languages could consider offering an English report (at least a short version) of their results on DS consumption.

The frequency of DS use varies significantly between countries, ranging from 5% in Italy to over 50% in Finland and Denmark. Firstly, this can be attributed to the different methodologies used. Inconsistent reference periods for DS consumption among studies may have influenced results. For example, in Belgium, DS users were 38.3% based on FFQ and 18.2% based on the 24 h recalls [26]. Studies that recorded DS use based on 24 h dietary recalls may have missed seasonal DS users. The results may have also been impacted by varying study durations, dissimilar age groups and different statistical analyses. It is also plausible that the outcomes could have been influenced by variations in health policies and recommendations, in addition to inter-country socioeconomic and cultural disparities. 

DS users are more likely to be female. The reason behind this difference may be that women tend to follow a healthier diet and lifestyle than men [43]. Moreover, women, depending on the age group, consume certain DSs, i.e., prenatal vitamins or calcium and vitamin D for postmenopausal women in the concern of osteoporosis [44]. Additionally, in most of the studies included, DS use increases with age. Older people are in danger of undernutrition due to various reasons, such as difficulties in chewing, malabsorption, sensory deficits and comorbidities [45,46]. Therefore, supplementation may help the older population meet the dietary requirements. In order to overcome deficiencies in vitamin B12, vitamin D and calcium, which frequently occur in this population, many countries recommend supplementation of the aforementioned vitamins and minerals for the elderly [47]. Surprisingly, older people do not consume DSs for nutritional reasons as a priority but to boost their immunity [48] and for “treatment” reasons [49], which is particularly related to the health of specific body organs [50]. Caution is needed for this population due to polypharmacy. In a study by Qato et al., a potentially significant drug–drug interaction was present in 15.1% of older individuals [51].

Studies have shown that DS users tend to be female, older, with a higher level of education, physically active, have a normal BMI, and have a healthier diet [49]. Very few studies [23,26,27] linked DS usage with lifestyle characteristics in our review, and thus no conclusion can be drawn. 

MVM seems to be the most preferred supplement in all countries. Fish oil supplements were the most used supplements in Sweden [32] and the UK [36]. Similar to the findings in Skeie et al. [52], fish oil supplements are frequently used in northern countries [22,32,36,37]. Vitamin D is the most favored supplement in Belgium and the Netherlands [25,26]. It seems that the preference of DS types is driven by the different cultural and lifestyle characteristics of each country. In most of the studies, the older population consumed a greater percentage of vitamin D and calcium supplements. This is in accordance with findings that supplementation with the aforementioned micronutrients may ameliorate bone health and prevent bone fractures in the elderly [53,54]. However, the total intake of vitamin D and calcium in some studies was lower than the recommended.

Few studies recorded the contribution of DSs to micronutrient intake [22,23,25,26,30,34,36,37]. The omission of the DS contribution to nutrient intake can be misleading since it poses the risk of underestimating total nutritional intake. As a result, comparisons between different populations and countries would be flawed [52]. Moreover, it is of great importance to clarify if DS users actually require DSs based on their diet. Interestingly, as it was shown in academic publications that recorded micronutrient intake, supplement usage was unnecessary or redundant since the nutritional needs were covered by diet alone for most of the vitamins and minerals. This phenomenon has been characterized before by Kirk et al. as the “inverse supplement hypothesis”, i.e., those who need supplements the least are oddly the ones who use them the most [55]. It is thus crucial to explore this tendency since the consumption of DSs by individuals who already meet daily DRVs through diet poses the risk of them exceeding UL, as was the case in Germany and Poland [30,34]. As a result, the percentage of people who consume micronutrients below the estimated average requirement (EAR) will not decrease, and at the same time, there may be a rise in the number of people at risk of exceeding the safe intake limits [10]. It is crucial to recognize populations with inadequate or excessive intakes and apply corresponding strategy plans. A study regarding the micronutrient intake of the Dutch population found that over 50% of the population presented consumption below the EAR concerning calcium, iron, vitamin D, and folate but simultaneously the UL for folic acid, vitamin B6, vitamin A, zinc, and iron, was surpassed by above 1% of some subgroups, especially children [56]. Flynn et al. found that supplements significantly boosted the nutrient consumption associated with the 95th centile of retinol, zinc, iodine, copper, and magnesium, exceeding the UL in a small portion of the populations under study [57]. In the French NutriNet-Santé cohort, DS use was found to cause excessive iron and magnesium intake, and potentially “threatened” DS use practices were reported [58]. There is a lack of large studies monitoring the possible adverse health effects of long-term excessive intake above the ULs [12,59]. Therefore, it is imperative to implement targeted monitoring and intervention programs of supplementation for the population subgroups that require them [59]. More precisely, whether the amount exceeds the population reference intake (PRI) or even the UL recently defined by EFSA for Europe [60,61] is to be examined for particular nutrients with no defined UL. Of note, there is no recommendation available. In addition, there is a significant concern regarding the population consuming supplements containing certain nutrients with no defined UL [60]. Their non-approved ingredient content (such as pharmaceuticals) is also debated [62]. 

Whether the DS intake contributes significantly to the nutrient status of an individual remains to be explored since bioavailability, synergistic/antagonistic mechanisms and other issues influence the absorption process in the human body [63,64]. Moreover, biomarkers need to be tested in national representative surveys to confirm their association with nutrition status and to better define biomarkers of inadequacy and overabundance of micronutrients [56,59]. 

The documented variations in DS consumption could trigger the scientific community and authorities to explore this issue in more detail to avoid potential misinterpretations. The recognition of vulnerable populations may promote the development of public health and nutritional policies regarding targeted populations at risk. Additionally, identifying the flaws relevant to DS documentation may improve policies linked to the evaluation of DS usage. The fact, as such, that there are variations is considered a starting point for clarifying the complicated DS consumption habits of the population in different countries in Europe. 

### 4.1. Public Health Issues on Dietary Supplement (DS) Intake

It is of great importance to record the reasons that lead individuals to consume DSs, something that is scarce in a nationally representative sample. Describing in detail the motives for consuming DS should be studied in depth, and the drivers in each European country should be identified in detail. This way, more targeted advice could be articulated on a national level.

Although according to EFSA [2], “Food supplements are intended to correct nutritional deficiencies, maintain an adequate intake of certain nutrients”, it has been documented in many surveys that DS users tend to consume DSs for health reasons rather than to fill nutrition gaps. Higher in their priority is to achieve or preserve a good health status [50,58,65]. In this example, 41.3% of the participants in a Polish study believed DSs were drugs [66]. Just one-quarter use DSs after advice from a healthcare professional [50]. 

To avoid the onset of chronic illnesses, more than one-third of adults in the US and Europe consume a MVM supplement daily [4]. Studies evaluating the need for DS intake have generated conflicting results. Some studies concluded that there are positive health effects of consuming a MVM daily [67]. Some other scientists showed that DSs have no impact on health [68], and other authors expressed susceptibility to consequent harm [7,12,69]. Randomized clinical trials exhibited a link between MVM and a reduction in all cancer incidence in men [70] as well as an association between antioxidant use to lower cancer incidence and all-cause mortality [71]. However, the recent COSMOS trial failed to exhibit a reduction in total cancer incidence in older MVM users [72]. Meta-analyses show that there is no correlation between MVM and cardiovascular disease (CVD), coronary heart disease (CHD) or stroke mortality [73]. In the VITAL study, supplementation with omega-3 fatty acids and vitamin D did not lessen the risk for cardiovascular events or cancer [13]. On the other hand, there are studies that have shown promising results, such as a possible link to decreased cancer incidence with MVM supplements [74] or supplementation with vitamin D linked with a lower risk of all-cause mortality [75]. Nevertheless, data are inadequate to recommend the use of DSs for the diminution of non-communicable diseases such as CVD events and cancer and all-cause mortality [12,76,77]. Indeed, several compounds have been shown to be harmful. Beta-carotene, vitamin E and high doses of vitamin A seem to increase mortality [78], while two RCTs showed that beta-carotene increased lung cancer in smokers [79,80]. Moreover, vitamin E increased prostate cancer in the SELECT study [81].

The combination of prolonged healthy nutrition habits with the consumption of DSs decreased the incidence of CVD [82]. Further studies are needed to explore the health impact DSs have according to diet habits. The NHANES study showed that a lower risk of mortality was linked only to sufficient food nutrients and not to the consumption of the same micronutrients via supplements [83]. The consumption of calcium above the ULs was related to augmented cancer mortality but only when it concerned supplements. On the other hand, an EPIC-Norfolk meta-analysis concluded that calcium supplementation might reduce mortality in women [84].

In addition, there is a significant concern regarding the population consuming supplements containing non-nutrients (e.g., herbal plants, botanicals, algae) where official PRIs and ULs are not defined [85]. The non-nutrient supplements, mainly of plant origin, are acceptable to the consumer [86,87], who assumes that they are safe and harmless [88]. There is a necessity for official guidelines regarding the appropriate use of nutritional supplements and herbal over-the-counter preparations [89].

Findings showed that supplement use is a reasonably unstable behavior in free-living individuals. The preferences for different DS types showed that local factors, such as culture and environment, may affect consumers’ decisions [90]. Adverse events of DSs are thought to be the cause of 23,000 emergency room visits annually in the United States [91], so caution is needed.

### 4.2. COVID-19 Pandemic and DS Use

The onset of COVID-19 focused consumers’ attention on solutions to boost their immune systems and combat the virus’s symptoms. Google trend analysis by Hamulka et al. revealed that interest in supplements and food elements such as vitamin C, vitamin D, turmeric and garlic peaked during the first two waves of the COVID-19 pandemic [92]. During the COVID-19 lockdown, the most used vitamins were vitamin C, vitamin D and zinc [93]. Several studies exhibited an increase in the use of DSs before and during COVID-19 in Europe [94,95]. There is inadequate evidence to recommend in favor of or against DSs for the prevention or therapy of COVID-19 [96]. Studies published have yielded conflicting results [97,98,99,100].

### 4.3. Strengths and Limitations

The current review’s strength is that it provides, for the first time in our knowledge, an up-to-date description of DS use at a national representative level among European countries. Several limitations of this review can be attributed to the varied methodologies employed in the surveys. A harmonized definition of a DS user is lacking. Thus, inter-country comparisons can be challenging. Additionally, although our search was meticulous, we are unable to guarantee that we identified all relevant NNS. Finally, the language barrier that was applied may have led to the exclusion of important, related data.

## 5. Conclusions

As it was shown, few published data exist regarding DS use in NNS. Although there are studies that have described the record of DS use in their methodology, published results are lacking. Data on DS consumption are vital for national nutrition monitoring. They can contribute to characterizing patterns of DS use in the population, estimating nutrient intake from supplements, evaluating total nutrient intake from food and DSs, and examining the diverse diet–health relationships. The best way to record DS consumption is still under discussion. In Europe, the creation and implementation of a standardized methodology remain necessary. EFSA, particularly, the EU menu project [101], is addressing issues of harmonizing methodologies for food consumption data, also using the Foode×2 system for DS categories. For adults, recommendations include the collection of data regarding DSs via 24 h recalls with the addition of FPQ, besides recording frequency and seasonality. Moreover, in this direction, EFSA analyzed the available information on food supplement consumption and created a database from which standard weights per type of supplement can be extracted [102]. Using a database to calculate the nutritional composition of DSs allows the studies to estimate the contribution of DSs to micronutrient intake and provide a better assessment of this intake between surveys [103]. The current study could contribute to developing a national infrastructure for monitoring the progress of specific targets supporting national and European policies and future interventions on diet. The current summarized state of the art expects to contribute to the development of national and European evidence-based policies that translate research into effective nutrition and health strategies, sustainable across time. 

## Figures and Tables

**Figure 1 nutrients-15-05090-f001:**
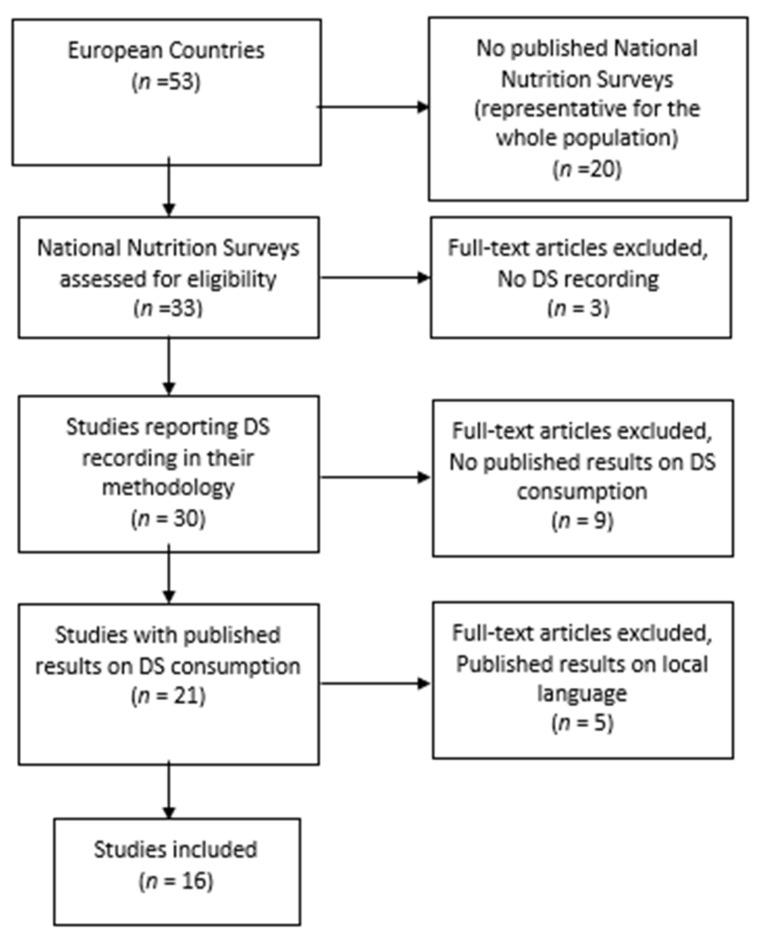
Flow diagram of study selection process.

**Table 1 nutrients-15-05090-t001:** Dietary supplement use examined in national nutrition surveys in 53 European countries.

Data on Dietary Supplements Available (*n* = 21)	Data on Dietary Supplements Not Performed or Unavailable (*n* = 26)	
Andorra	Azerbaijan	No data
Belgium	Albania	No data
Chech republic	Armenia	No data
Denmark	Austria	DS methods
Finland	Belarus	No data
France	Bosnia Herzegovina	No data
Germany	Bulgaria	No DS records
Greece	Croatia	No data
Hungary	Cyprus	DS methods
Iceland	Estonia	DS methods
Ireland	Georgia	No data
Italy	Kazakhstan	No data
Lithuania	Kyrgyzstan	No data
Netherlands	Latvia	DS methods
Norway	Licthtestein	No data
Poland	Luxemburg	No DS records
Portugal	Malta	No data
Spain	Moldova	DS methods/ DS results in certain groups
Sweden	Monaco	No data
Switzerland	Montenegro	DS methods
UK	North Macedonia	No data
	Romania	DS methods
	Russia	No DS records
	San Marino	No data
	Serbia	DS methods
	Slovakia	No data
	Slovenia	DS methods
	Tajikistan	No data
	Turkey	No data
	Turkmenistan	No data
	Uzbekistan	No data
	Ukraine	No data

**Table 2 nutrients-15-05090-t002:** National nutrition surveys citing dietary supplements in 16 European countries in the period 2000–2020.

Country	Name of Survey	Study Period	Number of Participants	Age Range, Years	Dietary Method	Reference
Belgium	Belgium National Food Consumption Survey (BNFCS) 2014	2014–2015	3146	3–64	FFQ, 2 × 24 h recall. Face-to-face electronic interview.	[26]
Denmark	Danish National Survey of Dietary Habits and Physical Activity	2000–2004	4479	18–75	Pre-coded 7day dietary record, personal interview.	[23]
Finland	The National FINDIET 2017 survey	2017	3099	18–74	2 × 24 h recall. Face-to-face interview and telephone.	[22]
France	INCA 3	2014–2015	5800 (3157 adults 2698 children)	0–79	Specific questionnaire face-to-face interview for a 12-month period for adults.	[31]
Germany	NVSII—German National Nutrition Survey (Nationale Verzehrstudie II)	2005–2007	13,753	14–80	2 × 24 h recall. Face-to-face electronic interview.	[30]
Greece	HYDRIA—Greek national diet and health survey	2013–2014	4011	>18	2 × 24 h recall; FPQ questionnaire. Face-to-face interview.	[27]
Hungary	Hungarian Dietary Survey 2009-OTAP2009	2009	4992 (3982 adults)	0–101	Targeted questionnaire on DS; 3-day diary, FFQ.	[29]
Ireland	National Adult Nutrition Survey 2011 (NANS)	2008–2010	1500	18–90	4-day semi-weighed food diary (consecutive). Self-completed.	[37]
Italy	INRAN-SCAI, The third Italian national food consumption survey	2005–2006	3323	0–98	3-day diary (consecutive). Self-completed.	[35]
Netherlands	DNFCS—Dutch National Food Consumption Survey 2012–2016	2012–2016	4313	1–79	2 × 24 h recall and 1-day food diary (some age groups), FFQ.	[25]
Poland	National Dietary cross-sectional Survey	2019–2020	1831 adults	>18	FPQ, 2 × 24 h recalls.	[34]
Portugal	IAN-AF 2015–2016—The National Food, Nutrition and Physical Activity Survey of the Portuguese general population	2015–2016	5811	3 months–84	2 × 24 h recall (non-consecutive) and FPQ (electronic interview), Face-to-face electronic interview.	[28]
Spain	ENALIA2	2013–2015	1033	18–74	FPQ, 2 × 24 h recalls	[33]
Sweden	Swedish National Dietary Survey—Riksmaten vuxna, 2010–2011	2010–2011	1797	18–80	4-day food diary (consecutive). Self-completed via web.	[32]
Switzerland	MenuCH—National Nutrition Survey menuCH 2014–2015 (Nationale Ernährungserhebung menuCH = Enquête nationale sur l’alimentation menuCH = Sondaggio nazionale sull’alimentazione menuCH)	2014–2015	2085	18–75	Self-administered dietary questionnaire. 2 × 24 HDR.	[24]
UK	National diet and Nutrition Survey (NDNS)	2008/09–2011/12	6828 (3450 adults)	>1.5	4-day food diary. Face-to-face interview.	[36]

**Table 3 nutrients-15-05090-t003:** Dietary Supplement users, percentages by gender, and total (%).

Country	Men (%)	Women (%)	Total (%)
Belgium	29.1/12.3 *	47/23.8 *	38.3/18.2 *
Denmark	51	60	no data available
Finland	52	64	no data available
France	17	26.3	21.8
Germany	19	29.6	24.3
Greece	21.8	39.9	31.2
Hungary	22.6	27.8	25.5
Ireland	22	33	28
Italy	3/6 ^¥^	5/8 ^¥^	5
Netherlands	30	52	42
Poland	6	13	10
Portugal	21.5	31.5	26.6
Spain	no data available	no data available	13.3
Sweden	15	27	21
Switzerland	38.1	56.4	47
Uk	17/27 ^§^	35/47 ^§^	22/41 ^§^

* based on FFQ/based on 24 h recall. ^¥^ 18–65 years old/≥65 years old. ^§^ 19–64 years old/≥65 years old.

## Data Availability

Not applicable.

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
