# Peer review of "Is Dietary (Food) Supplement Intake Reported in European National Nutrition Surveys?"

_nutrients, 2023, doi:10.3390/nu15245090_

Round 1

Reviewer 1 Report

Comments and Suggestions for Authors

Thank you for this excellent and very thoughtful and thorough review. This is an important topic for all of the reasons you have mentioned - especially that if nutrient DS use is not estimated as part of surveys of national nutrient intake, the understanding of national nutrient status is inaccurate.  

This reviewer has only a few small edits to recommend:

Comments: 

You appear to be using the abbreviation DS as both singular and plural throughout the manuscript. I recommend that you define the DS abbreviation this way in the beginning of the manuscript or use DSs for the plural version.

line 32: at the end of the first sentence, I recommend that you include the DSHEA definition of DS for the reader's comparison with the EFSA definition.

line 40: please add: ".. as medicine or DS in another."

line 159: Please delete the word "likings" and replace it with the word "preferences"

Line 236: Please replace header with "Contributions of DS to the Diet"

 Thank you again for this excellent manuscript!

Reviewer 2 Report

Comments and Suggestions for Authors

This manuscript addresses the increasing use of dietary supplements (DS) and the importance of understanding this trend for national nutrition monitoring in European countries. While the topic is relevant and the study provides valuable insights, there are some aspects that could be considered and potential criticisms to be addressed:

1. Methodology and Data Collection:

   - The manuscript mentions retrieving data from various sources, including literature review, scientific publications, and open-published reports. A more detailed explanation of the search strategy, inclusion/exclusion criteria, and potential biases in data collection would strengthen the methodology. 

2. Language Limitation:

   - The inclusion criteria specify publications in English, French, or German. This may introduce a language bias, potentially excluding relevant data published in other languages. The rationale for this limitation should be clearly stated, and potential consequences discussed.

3. Representativeness of Data:

   - The study reports data from 30 out of 47 European countries, and findings from only 21 cases include percentages of DS intake. The representativeness of the included countries and the impact of missing data on the overall conclusions should be acknowledged and discussed.

4. Data Variability:

   - The manuscript notes significant variability in DS use among European countries. While this information is valuable, it would be beneficial to explore potential reasons for such variations, such as cultural factors, healthcare practices, or socioeconomic differences. 

5. Quality and Consistency of Reporting:

   - The manuscript highlights a need for improvement in comprehensive data reporting on DS consumption. Elaborating on the specific deficiencies in the current reporting systems and suggesting potential strategies for standardization would strengthen the paper.

6. Implications for Policy and Practice:

   - The manuscript could benefit from a discussion on the practical implications of the findings. How can the observed variations in DS use inform public health policies and interventions? What are the potential risks and benefits associated with these variations?

 7. Future Research Directions:

   - The conclusion could be enhanced by suggesting specific areas for future research. For instance, proposing studies to explore the reasons behind the observed differences in DS use or recommending strategies for harmonizing DS categorization across surveys.

 8. Acknowledgment of Limitations:

   - Explicitly acknowledging the limitations of the study, such as potential underreporting, data inaccuracies, or the lack of standardized definitions for DS categories, would provide a more transparent interpretation of the results.

 Addressing these potential criticisms and incorporating suggestions for improvement would contribute to the overall strength and impact of the manuscript.

Comments on the Quality of English Language

 Adjusting sentence structures and providing additional clarity in certain statements can further enhance the readability of the text.
